# Rational design of aptazyme riboswitches for efficient control of gene expression in mammalian cells

**Guocai Zhong[1]\*, Haimin Wang[1], Charles C Bailey[2], Guangping Gao[3], Michael Farzan[1]**

[1]Department of Immunology and Microbial Sciences, The Scripps Research Institute, Jupiter, United States; [2]Department of Molecular and Comparative Pathology, Johns Hopkins School of Medicine, Baltimore, United States; [3]Gene Therapy Center, University of Massachusetts Medical School, Worcester, United States

**Abstract** Efforts to control mammalian gene expression with ligand-responsive riboswitches have been hindered by lack of a general method for generating efficient switches in mammalian systems. Here we describe a rational-design approach that enables rapid development of efficient *cis*-acting aptazyme riboswitches. We identified communication-module characteristics associated with aptazyme functionality through analysis of a 32-aptazyme test panel. We then developed a scoring system that predicts an aptazymes's activity by integrating three characteristics of communication-module bases: hydrogen bonding, base stacking, and distance to the enzymatic core. We validated the power and generality of this approach by designing aptazymes responsive to three distinct ligands, each with markedly wider dynamic ranges than any previously reported. These aptayzmes efficiently regulated adeno-associated virus (AAV)-vectored transgene expression in cultured mammalian cells and mice, highlighting one application of these broadly usable regulatory switches. Our approach enables efficient, protein-independent control of gene expression by a range of small molecules.

\*For correspondence: gzhong@ scripps.edu

**Competing interests:** The authors declare that no competing interests exist.

## Introduction

Ligand-responsive *cis*-acting riboswitches (*Breaker, 2012*) function independently of protein factors and can be programmed to respond to a wide range of small-molecule ligands. They therefore have a number of potential scientific and medical applications. Aptazymes are a class of engineered riboswitches that control gene expression by small molecule-induced self-cleavage of a ribozyme (*Figure 1A*). The first aptazyme was developed by the Breaker group (*Tang and Breaker, 1997*) by fusing an ATP binding aptamer with a minimal hammerhead ribozyme (*Scott et al., 2013*). This aptazyme was functional only in cell-free systems. The discovery of tertiary interaction between stems I and II of the full-length hammerhead ribozyme (*De la Peña et al., 2003*; *Khvorova et al., 2003*; *Martick and Scott, 2006*) and the development of optimized hammerhead ribozymes efficient in mammalian cell culture and in vivo (*Yen et al., 2004*) led to the development of hammerhead ribozyme-based aptazymes functional in yeast (*Win and Smolke, 2007*; *Wittmann and Suess, 2011*; *Klauser et al., 2015*; *Townshend et al., 2015*), or mammalian cells (*Kumar et al., 2009*; *Auslander et al., 2010*; *Nomura et al., 2012*; *Wei et al., 2013*; *Beilstein et al., 2015*). These aptazymes have been tested for a range of applications, including construction of synthetic gene networks in yeast (*Win and Smolke, 2008*), and regulation of transgene expression (*Ketzer et al., 2012*) or virus replication (*Ketzer et al., 2014*) in mammalian cell culture. Using IL-2-based positive-feedback signal amplification, a theophylline-regulated aptazyme with a narrow dynamic range has

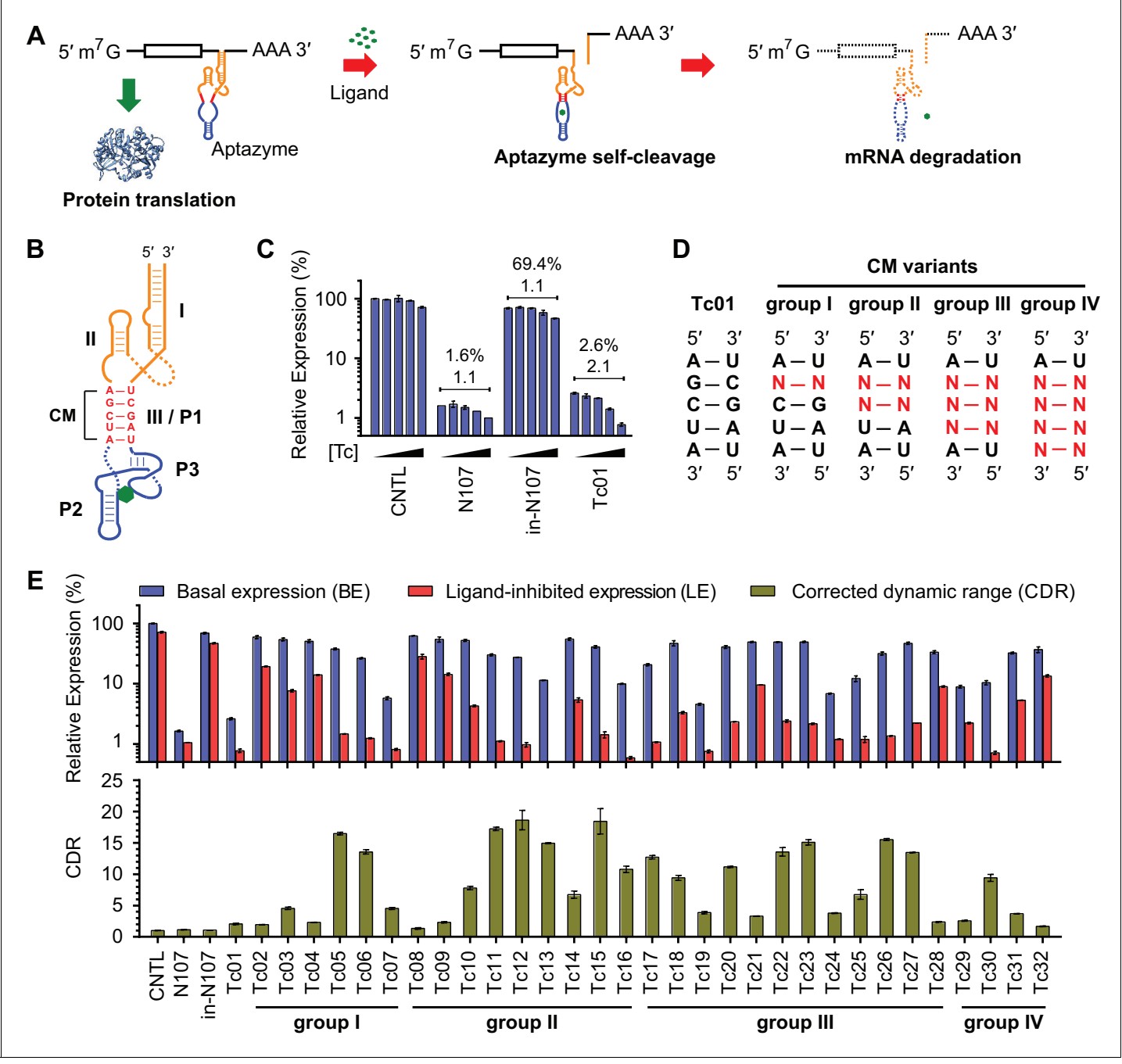

**Figure 1.** A test panel of 32 tetracycline (tc) aptazymes. (**A**) A diagram representing aptazyme-mediated control of gene expression. An aptazyme riboswitch, composed of a ribozyme (orange), a communication module (red) and an aptamer (blue), is inserted at the 3' untranslated region of an mRNA transcript. Ligand binding to the aptamer activates the ribozyme, which cleaves itself in *cis*, leading to degradation of the RNA message. (**B**) Secondary structure of the Tc01 aptazyme, composed of a hammerhead ribozyme (orange), a communication module (CM, red), and a Tc-binding aptamer (blue). Tc is represented as a green hexagon. The original stem III of the hammerhead ribozyme and P1 stem of the Tc aptamer are replaced with the communication module, which connects the ribozyme and aptamer and signals the binding state of the aptamer to the ribozyme. (**C**) Enzymatically active (N107) and inactive (in-N107) forms of the N107 hammerhead ribozyme, or the aptazyme Tc01 were placed 3´ to a *Gaussia* luciferase (GLuc) gene. HeLa cells transiently transfected with plasmids expressing GLuc regulated by N107 or Tc01 were cultured with 0, 3, 10, 30, or 100 µM Tc for 2 days. Secreted GLuc in the culture supernatant was measured by a luminescence assay. Luciferase expression levels are normalized to that of an expression construct ('CNTL') lacking any regulatory elements. The percentages above the figure indicate 'basal expression' (BE), GLuc expression from each construct in the absence of Tc relative to that from the CNTL control construct. The lower number above each figure indicates the 'corrected dynamic range' (CDR) at 100 µM Tc. CDR is the Tc-induced fold-change in aptazyme-regulated GLuc expression divided by the Tc-

*Figure 1 continued on next page*

*Figure 1 continued*

induced fold-change in GLuc expression from the CNTL control construct. (D) To gain insight into the relationship between communication-module sequence and CDR, 32 aptazyme variants, differing only in their communication modules and represented here in four groups, were generated. Full communication-module sequences are shown in *Figure 1—figure supplement 1B*. (E) HeLa cells transiently transfected with plasmids expressing GLuc regulated by these 32 aptazyme variants were cultured in the presence or absence of 100 μM Tc for 2 days. Secreted GLuc in the culture supernatant was analyzed as in (C). The upper panel shows the BE (blue) and 'ligand-inhibited expression' (LE; red) of each variant. LE is the relative luciferase expression of each plasmid in the presence of Tc compared to the luciferase expression level of the CNTL control plasmid in the absence of Tc. The lower panel shows the CDR of each variant. All data shown are representative of two or three independent experiments. All data points represent mean ± S.D. of three biological replicates.

The following figure supplements are available for figure 1:

**Figure supplement 1.** Sequence and secondary structure of test-panel Tc aptazymes.

**Figure supplement 2.** Characterization of test-panel Tc aptazymes.

been shown to control proliferation of transduced T cell lines in mice (*Chen et al., 2010*). To date, no aptazyme has been described to control gene expression in primary animal cells in vivo.

The limited number of useful aptazymes reflects the need for a simple and general strategy to rapidly generate aptazymes responsive to diverse ligands in mammalian cells. In vitro allosteric selection has been used to generate aptazymes in a high-throughput and massively parallel format (*Koizumi et al., 1999*), but aptazymes identified in this way function in cell-free conditions but not in mammalian cells (*Link et al., 2007*; *Wittmann and Suess, 2011*). Similarly, aptazymes obtained from bacterial screens function efficiently in bacteria (*Wieland and Hartig, 2008*) but poorly in mammalian cells (*Auslander et al., 2010*). Direct screening of an aptazyme library in mammalian cells is labor intensive and generally performed in a medium-throughput non-pooled format (*Auslander et al., 2010*; *Rehm et al., 2015*), thus library sizes are small, and the chance of identifying an efficient aptazyme is low.

In this study, we describe a general method for rapid development of efficient aptazymes from diverse pre-existing or novel aptamers. Specifically, this approach streamlines identification of optimal communication modules connecting an aptamer to a hammerhead ribozyme. We demonstrate the power and modularity of this approach by developing efficient aptazymes regulated by three distinct ligands, each aptazyme exhibiting a significantly wider dynamic range in mammalian cells than any previously described aptazyme. We highlight the utility of these aptazymes by using them to regulate expression from an AAV vector in cell culture and in mice.

## Results

### Characterization of a test panel of tetracycline-regulated aptazymes

We initially characterized a panel of aptazymes responsive to tetracycline (Tc). These aptazymes share a common architecture, composed of an optimized *Schistosoma mansoni* hammerhead ribozyme N107 (*Yen et al., 2004*), a communication module, and a Tc-binding aptamer (*Xiao et al., 2008*) (*Figure 1—figure supplement 1A*). They differ only in their communication module, a region that signals the binding state of the aptamer to the ribozyme. We started by directly fusing four base pairs of the P1 stem of the aptamer to the base of ribozyme stem III (*Figure 1B*). The resulting aptazyme (Tc01) was placed downstream of a gene encoding *Gaussia* luciferase (GLuc). As expected, Tc01-regulated GLuc expression in the absence of Tc was markedly lower than that from control constructs without any regulatory elements (CNTL) or with an inactive form of the N107 ribozyme (in-N107), likely due to high level of ribozyme auto-activation triggered by its stable communication module. In addition, the aptazyme had a narrow (2.1-fold) dynamic range (*Figure 1C*). We then generated 31 variants of Tc01 by altering Tc01 communication-module bases by trial and error (*Figure 1D* and *Figure 1—figure supplement 1B*). These variants were again placed downstream of the GLuc gene and tested for their ability to control GLuc expression in HeLa cells using varying concentrations of Tc (*Figure 1—figure supplement 2*). A comparison of GLuc expression at 0 μM Tc

(basal expression, BE) and GLuc expression at 100 µM Tc (ligand-inhibited expression, LE) is plotted in *Figure 1E*. As shown, this test panel exhibited widely varying BE and the Tc-induced fold change in expression. To correct for the non-specific inhibitory effects of Tc on gene expression, we report here the corrected dynamic range (CDR), namely the Tc-induced fold change in aptazyme-regulated gene expression divided by the fold change observed in the expression of a control construct (CNTL) lacking any regulatory elements.

## A function of communication-module bases that predicts an aptazyme's dynamic range

To better understand the determinants of aptazyme dynamic range, we comprehensively analyzed the communication-module characteristics and the functionality of the test-panel 32 aptazymes. It has been suggested that the annealing energy of the communication module, $\Delta G_{CM}$, determines CDR (*Wieland and Hartig, 2008*; *Kumar et al., 2009*). However, we observed no obvious relationship between predicted $\Delta G_{CM}$ (*Bellaousov et al., 2013*) and BE, LE, or CDR (*Figure 2A* and *Figure 2—figure supplement 1A*). We then focused on BE because it reflects the efficiency of ribozyme-mediated mRNA cleavage in the absence of ligand and is primarily determined by the communication module (*Figure 2—figure supplement 2A*). We hypothesized that an energy-like function of the communication-module sequence which predicts BE would identify values of the function where a small change would have the largest impact on target gene expression. Thus, for communication modules with these values, additional energy provided by aptamer binding would yield the largest differences between BE and LE, and thus the largest CDR. We therefore sought to construct a simple function that correlates with the rank order of BE among our 32 aptazymes.

We observed, in two different sets of aptazymes with identical communication-module base pairs, that distance of the base pairs to the enzymatic core impacted BE (*Figure 2B and C*). In particular, when base pairs with few hydrogen bonds (G-U or U-U) were placed close to the ribozyme, as in Tc14 and Tc08, higher BE levels were observed. Accordingly, we assigned every base a weight reflecting its proximity to the enzymatic core, and multiplied that weight by the number of hydrogen bonds to get a Weighted Hydrogen Bond Score (WHBS). We found that WHBS better correlates with the rank order of BE than does $\Delta G_{CM}$ or the unweighted sum of hydrogen bonds in the communication module (*Figure 2D* and *Figure 2—figure supplement 1B–E*).

Inter-strand purine base stacking (here referred to as 'base stacking') can also contribute to the stability of A-form RNA duplexes (*Egli, 2009*). We observed in aptazymes with identical WHBS that base stacking in the communication module also affected BE in a position-dependent manner (*Figure 2E and F*). We thus amended WHBS to include a correction for base stacking. The resulting Weighted Hydrogen-bond and Stacking Score (WHSS) better correlates with rank order of aptazyme BE values than does WHBS or $\Delta G_{CM}$ (*Figure 2G* and *Figure 2—figure supplement 1C–F*). Notably, there is an inflection point in the BE values plotted in *Figure 2G* at WHSS of 6.7. At this value, the additional energy provided by the ligand-bound aptamer will have a large impact on the activation of the ribozyme and thus the aptazyme's dynamic range (*Figure 2—figure supplement 2B–D*). Indeed, when CDR is plotted against WHSS (*Figure 2H*), the highest CDRs were observed at WHSS values 6.75 (Tc15) and 6.92 (Tc12). Accordingly, we focused hereafter on communication-modules with WHSS values within the range of 6.7 ± 0.3.

We also noted in this analysis two outliers, Tc31 and Tc32, both of which had CDR scores substantially lower than what their WHSS values would predict. This effect is due to a high LE rather than a lower-than-expected BE. Comparison of Tc31 and Tc32 with Tc11 indicates that the aptamer-proximal base pair accounts for their low CDR values (*Figure 2I*). Consistent with this, aptamers with WHSS values similar to Tc31 and Tc32, but with an aptamer-proximal A-U base pair, had higher CDR values (*Figure 2J and K*). Because a communication module with WHSS value around 6.7 tends to have lower annealing energy than the P1 stem of the original Tc aptamer, we speculated that an aptamer-proximal base pair with few hydrogen bonds lowers the affinity of the aptamer for its ligand. In addition, proximity of this base pair to the aptamer may subtly impair aptamer folding. We therefore used the base pair present in the original aptamer, or one with higher stability, in our subsequent efforts. When this final *ad hoc* principle was incorporated, aptazymes with WHSS values 6.7 ± 0.3 had consistently higher CDRs than aptazymes outside of this range (*Figure 2L*).

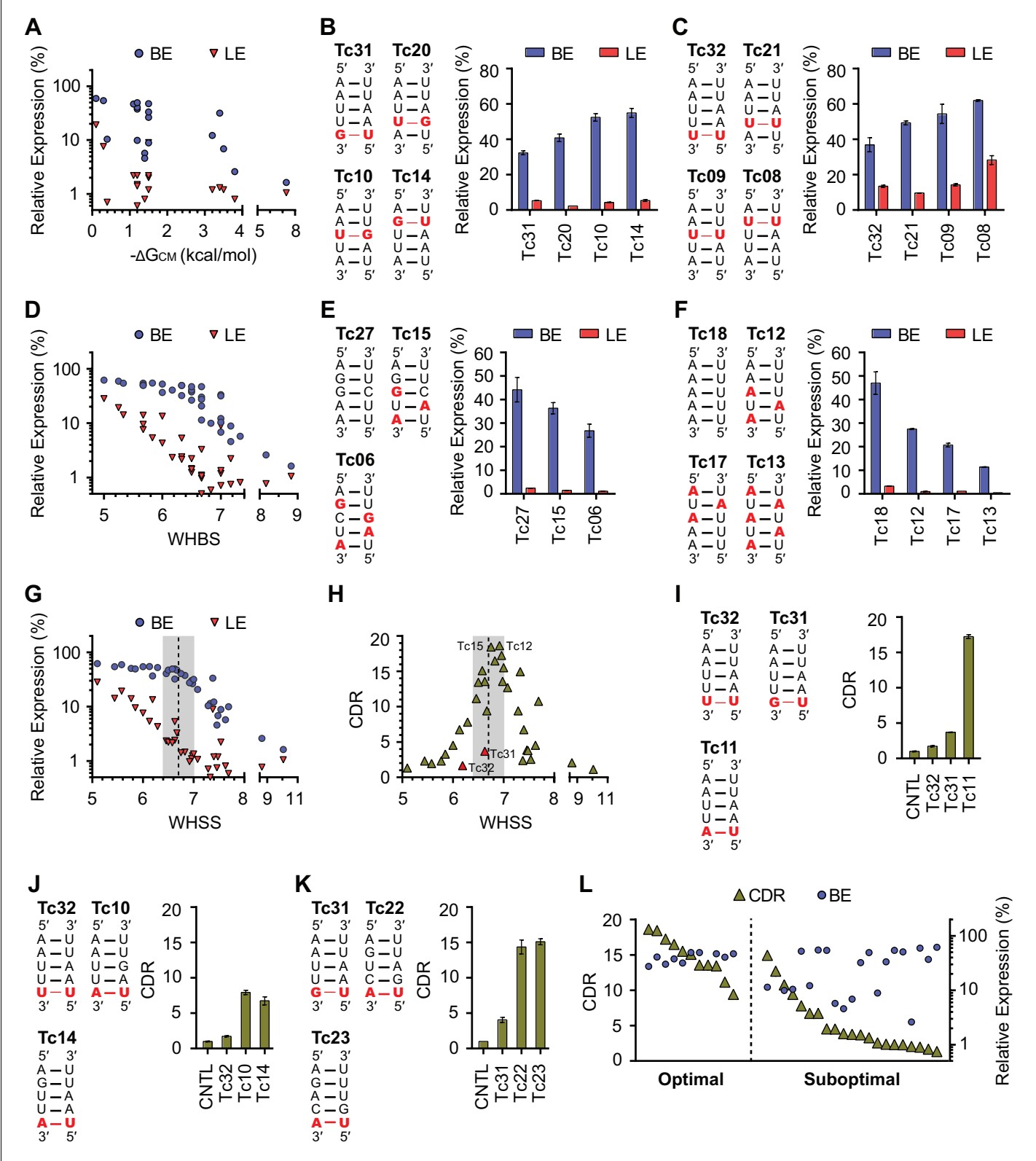

**Figure 2.** Development of rational-design principles. (**A**) The predicted hybrid free energies of the communication modules ($\Delta G_{CM}$) of test-panel aptazymes were calculated with RNAstructure Web Server (*Bellaousov et al., 2013*). BE (blue circles) and LE (red triangles) were then plotted against the negative $\Delta G_{CM}$ value for all aptazymes whose $\Delta G_{CM}$ could be determined. Note the poor correlation between $\Delta G_{CM}$ and BE or LE. (**B–C**) BE and LE of the indicated aptazyme variants are shown. Expression levels are normalized to that of the CNTL control construct. The communication-module

*Figure 2 continued on next page*

*Figure 2 continued*

sequences of these variants are displayed to the left of the figures. Red indicates base pairs different from the consensus. Note that all constructs in each set have the same base pairs, and that order of these pairs impacts BE. (D) Accordingly, every CM base pair was assigned a weight based on its proximity to the ribozyme, with distal bases assigned lower weights. A Weighted Hydrogen Bond Score (WHBS) was then calculated as the weighted sum of hydrogen bonds in the communication module. BE (blue circles) and LE (red triangles) of each variant were plotted against WHBS. Note that WHBS better correlates with the rank order of BE than does $\Delta G_{CM}$ or the unweighted sum of hydrogen bond numbers in the communication module (*Figure 2—figure supplement 1B*). (E–F) The BE and LE of the indicated aptazymes are shown, with their communication-module sequences indicated to the left. Nucleotides with the potential to form inter-strand base-stacking interactions are highlighted as red. (G) Potential inter-strand purine base-stacking interactions were weighted by proximity to the ribozyme, and added to the WHBS to form a modified score (WHSS). BE (blue circles) and LE (red triangles) were plotted against WHSS of the corresponding aptazyme variant. Note that the WHSS better correlates with the rank order of BE than does the WHBS or $\Delta G_{CM}$. An inflection point of the BE at WHSS 6.7 is indicated with a dashed line. Shading indicates a region with WHSS values 6.7 ± 0.3. (H) The CDR values of all 32 test-panel aptazymes were plotted against their WHSS. Note the CDR optimum near the WHSS value of 6.7 (shaded). Aptazymes that exhibited the highest (Tc12, Tc15) or lower-than-expected (Tc31 and Tc32) CDRs are indicated. (I–K) The CDRs of the indicated aptazymes are shown, with their communication-module sequences displayed to the left. Note that aptazymes in each panel share identical sequences (I) or similar WHSS (J,K), but differ in the stability of the communication-module base pair immediately proximal to the Tc aptamer (red). Aptazyme WHSS values range from 6.13 to 6.29 in (J), and from 6.58 to 6.63 in (K). Tc31 and Tc32, with 1 and 0 hydrogen bonds at this position, are outliers in *Figure 2H*, likely because these weak bonds destabilize the aptamer, and lower its affinity for Tc. (L) Efficient aptazymes met two criteria: a communication-module WHSS value within the range of 6.7 ± 0.3, and an aptamer-proximal communication-module base pair with stability similar to or higher than that of the original aptamer. Aptazymes that meet both criteria have high BE and CDR ('optimal' aptazymes), whereas those that fail to meet one have low BE or narrow CDR ('suboptimal' aptazymes). The CDR and BE of the both optimal and suboptimal aptazymes are ordered by CDR and plotted. Data points in panels A, D, G, H and L represent mean of three biological replicates. Data points in panels B, C, E, F, I, J, and K represent mean ± S.D. of three biological replicates. Numerical data for all the figures are available in *Figure 2—source data 1*.

The following source data and figure supplements are available for figure 2:

**Source data 1.** Communication-module scores and expression values of test-panel Tc aptazymes.

**Figure supplement 1.** Correlation analysis of energy-like functions describing the communication modules.

**Figure supplement 2.** The relationship between aptazyme activation and dynamic range.

## Validation of design principles through development of four classes of efficient aptazymes

To test the utility of the basic design principles, we first generated Tc aptazymes with the same architecture as the test panel, but with 4 bp, 6 bp, or 7 bp communication modules (*Figure 3A* and *Figure 3—figure supplement 1A*). All of these new aptazymes had WHSS values between 6.3 and 7.0, and their CDR values were all greater than 10 fold at 100 µM Tc (*Figure 3B* and *Figure 3—figure supplement 1B*), indicating that our rational approach is largely independent of communication-module length.

We then sought to determine if this basic heuristic was general to a wider range of aptazymes. We evaluated three more validation panels: (1) aptazymes using the same tetracycline aptamer, but fused at a different aptamer stem (Tc-P2 aptazymes), (2) aptazymes constructed using a theophylline aptamer (Theo aptazymes), and (3) aptazymes constructed with a guanine aptamer (Gua aptazymes). To date, all reported Tc aptazymes are fused at the aptamer P1 stem (*Win and Smolke, 2007*; *Wittmann and Suess, 2011*; *Beilstein et al., 2015*), but the structure of the Tc aptamer suggests that Tc-binding might have a larger effect on the P2 stem than the P1 stem (*Xiao et al., 2008*). We therefore explored variants of our Tc aptazymes in which the ribozyme was fused to the P2 stem of the Tc aptamer (Tc-P2 aptazymes; *Figure 3C*) rather than the P1 stem (Tc-P1 aptazymes; *Figure 3A*). All aptazymes except Tc39 and Tc44 exhibited high CDR (CDR ≥ 10 ± 0.5, *Figure 3D* and *Figure 3—figure supplement 1C*). Tc39 had a relatively low WHSS of 6.0, and Tc44 was intentionally designed with an aptamer-proximal A-U replacing the original G-C pair. Thus the basic principles developed with Tc-P1 aptazymes extend to a novel Tc-P2 aptazyme architecture. One of the Tc aptazymes with this alternative architecture, Tc40, outperformed all Tc-P1 aptazymes, and was used in subsequent studies. We then generated aptazymes constructed from entirely different aptamers, specifically those binding theophylline (*Jenison et al., 1994*) and guanine (Gua)

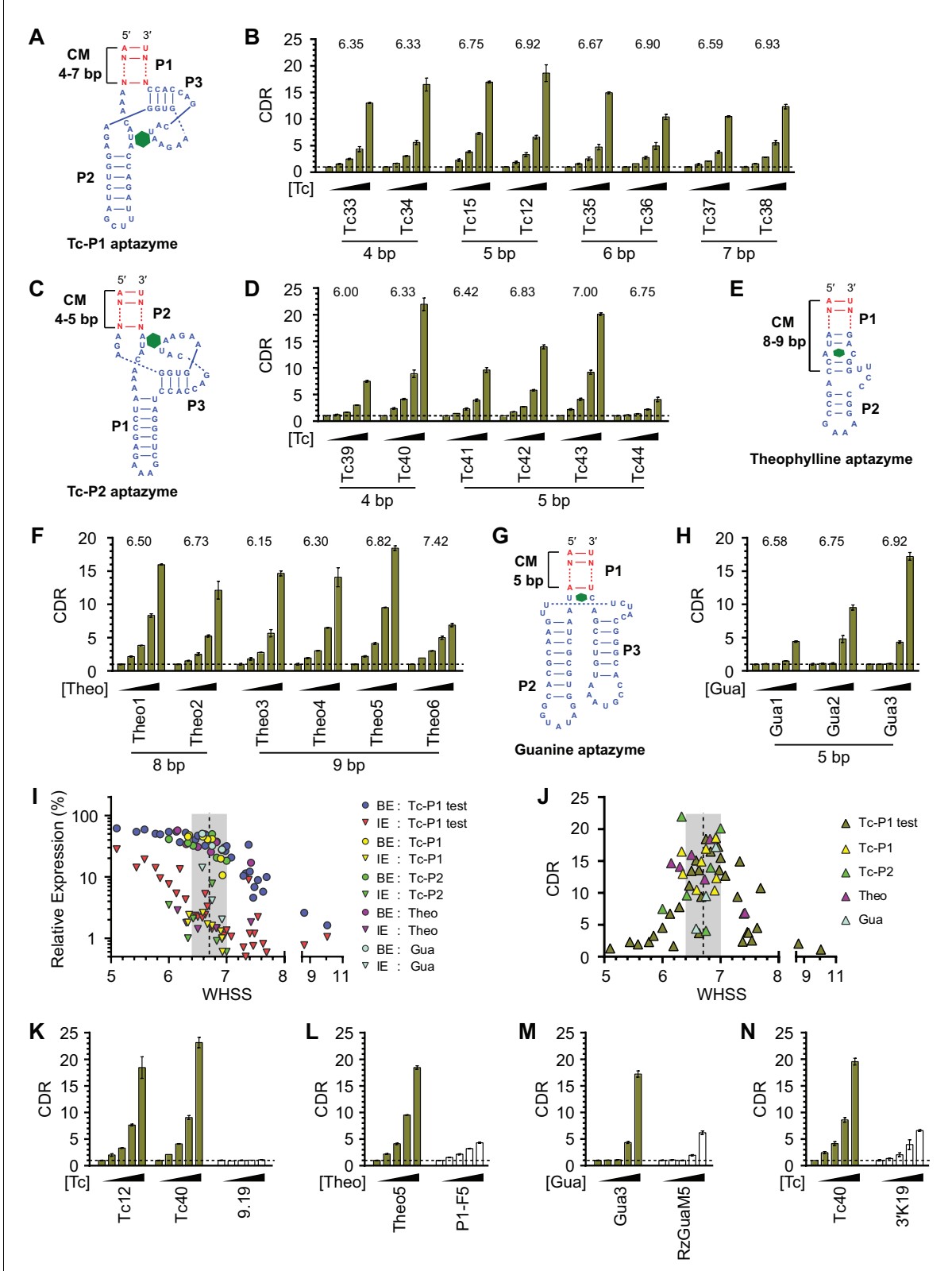

**Figure 3.** Validation of the design principles. (A–D) Secondary structures of Tc-P1 (A) or Tc-P2 (C) aptazymes, varying in communication-module lengths and the stem to which the ribozyme is attached, are shown without their ribozyme domains. HeLa cells transiently transfected with plasmids expressing GLuc regulated by Tc-P1 (B) or Tc-P2 (D) aptazymes were cultured with 0, 3, 10, 30, or 100 µM of Tc for 2 days. GLuc secreted into the culture supernatant was measured by a luminescence assay, and CDRs were calculated as in *Figure 1E*. WHSS values for each aptazyme are shown above

*Figure 3 continued on next page*

*Figure 3 continued*

figures. () Secondary structure of a theophylline (Theo) aptazyme, with its ribozyme domain omitted. (F) Experiments similar to (B) except that HeLa cells transiently transfected with plasmids expressing GLuc regulated by Theo aptazymes were cultured with 0, 0.1, 0.3, 1.0, 3.0 mM theophylline for 2 days. WHSS values are shown above figure. (G) Secondary structure of a guanine (Gua) aptazyme, with ribozyme omitted. (H) Experiments similar to (B) except that 293T cells transiently transfected with plasmids expressing GLuc regulated by Gua aptazymes were cultured with 0, 7, 20, 60, or 180 µM of guanine for 1 day. WHSS values are shown above figure. BE, LE, and CDR values of all validation-panel aptazymes are available in *Figure 3—source data 1*. (I) BE and LE of all aptazyme variants from the test panel and the four validation panels were plotted against WHSS of the corresponding variant. (J) CDRs of all aptazymes were similarly plotted against their WHSS values. (K–M) Aptazymes with the highest CDRs from each class (olive) are compared to previously described aptazyme off-switches (white) with the highest reported dynamic ranges. Tc-P1 aptazyme Tc12 and Tc-P2 aptazyme Tc40 are compared to 9.19 (*Wittmann and Suess, 2011*) (K); Theo5 is compared to P1-F5 (*Auslander et al., 2010*) (L); and Gua3 aptazyme is compared to RzGuaM5 (*Nomura et al., 2012*) (M). (N) Tc40 is compared to a previously described aptazyme on-switch, 3'K19 (*Beilstein et al., 2015*). Data shown are representative of two or three independent experiments. Data points in panels I and J represent mean of three biological replicates, and data points in remaining figures represent mean ± S.D. of three biological replicates.

The following source data and figure supplements are available for figure 3:

**Source data 1.** Communication-module scores and expression values of validation-panel aptazymes.
**Figure supplement 1.** Characterization of four validation panels of aptazymes.
**Figure supplement 2.** Comparison of our best Tc, Theo, and Gua aptazymes with previously described aptazyme off- or on-switches.

(*Mandal et al., 2003*) (*Figure 3E–H* and *Figure 3—figure supplement 1A,D, and E*). In both cases, we could readily generate aptazymes of high CDR, establishing the modularity of our approach.

*Figure 3I* plots the BE and LE of all the aptazymes against their WHSS values. Despite the diversity of these aptazymes, WHSS well predicted BE, consistent with our use of a common ribozyme, and the generality of our approach. Similarly, all aptazymes with high CDR had WHSS values near the region (WHSS 6.7 ± 0.3; grey) found to be optimal in *Figure 2H* (*Figure 3J*). Finally, we directly compared our best Tc, Theo, and Gua aptazymes with previously described aptazyme off-switches with the highest reported dynamic ranges. Consistent with the original reports, Tc aptazyme 9.19 (*Wittmann and Suess, 2011*) showed no functionality in mammalian cells (*Figure 3K*), while both P1-F5 (*Auslander et al., 2010*) and RzGuaM5 (*Nomura et al., 2012*) showed a maximal dynamic range of approximately 5-fold (*Figure 3L and M*). In each case, our rationally designed aptazymes substantially outperformed these previously described aptazymes (*Figure 3K–M*, and *Figure 3—figure supplement 2A–C*). We also directly compared our Tc40 off-switch with a previously described Tc aptazyme on-switch, 3'K19 (*Beilstein et al., 2015*). Again Tc40 showed significantly higher CDR (19.6-fold at 100 µM Tc) than the CDR (6.6-fold at 100 µM Tc) of this previously described aptazyme (*Figure 3N*, and *Figure 3—figure supplement 2D*).

## Aptazyme-mediated control of transgene expression in mammalian cell culture and in vivo

As shown in *Figure 3D*, Tc40 exhibited the highest CDR of all Tc-regulated aptazymes. We generated a Tc40 variant, Tc45, in which ribozyme N107 is replaced by the closely related hammerhead ribozyme N117 (*Yen et al., 2004*). Tc40 or Tc45 were then inserted either upstream (5') or downstream (3') of the GLuc gene, as a single copy (Tc40-5', Tc40-3', Tc45-3'), in tandem repeats (Tc40-5'5', Tc45-3'3'), or across the GLuc gene (Tc40×45) (*Figure 4A*). We observed in transiently transfected cells that Tc45-3' has a lower CDR than Tc40-3', but a higher BE, advantageous when aptazymes are used in tandem (*Figure 4B*). When these aptazymes were combined, with Tc40 placed 5' of the GLuc reporter and Tc45 placed 3', the resulting construct (Tc40×45) provided a 44-fold CDR with a 41% BE. We then measured regulation of GLuc in cells generated by retroviral transduction. Again, Tc40×45 exhibited a wider dynamic range than either Tc40-3' or Tc45-3' (*Figure 4C*) and the regulation observed was completely reversed when Tc was withdrawn (*Figure 4D*). Tc40×45 also showed efficient regulation when GLuc was replaced with a destabilized green fluorescent protein (*Figure 4E* and *Figure 4—figure supplement 1*).

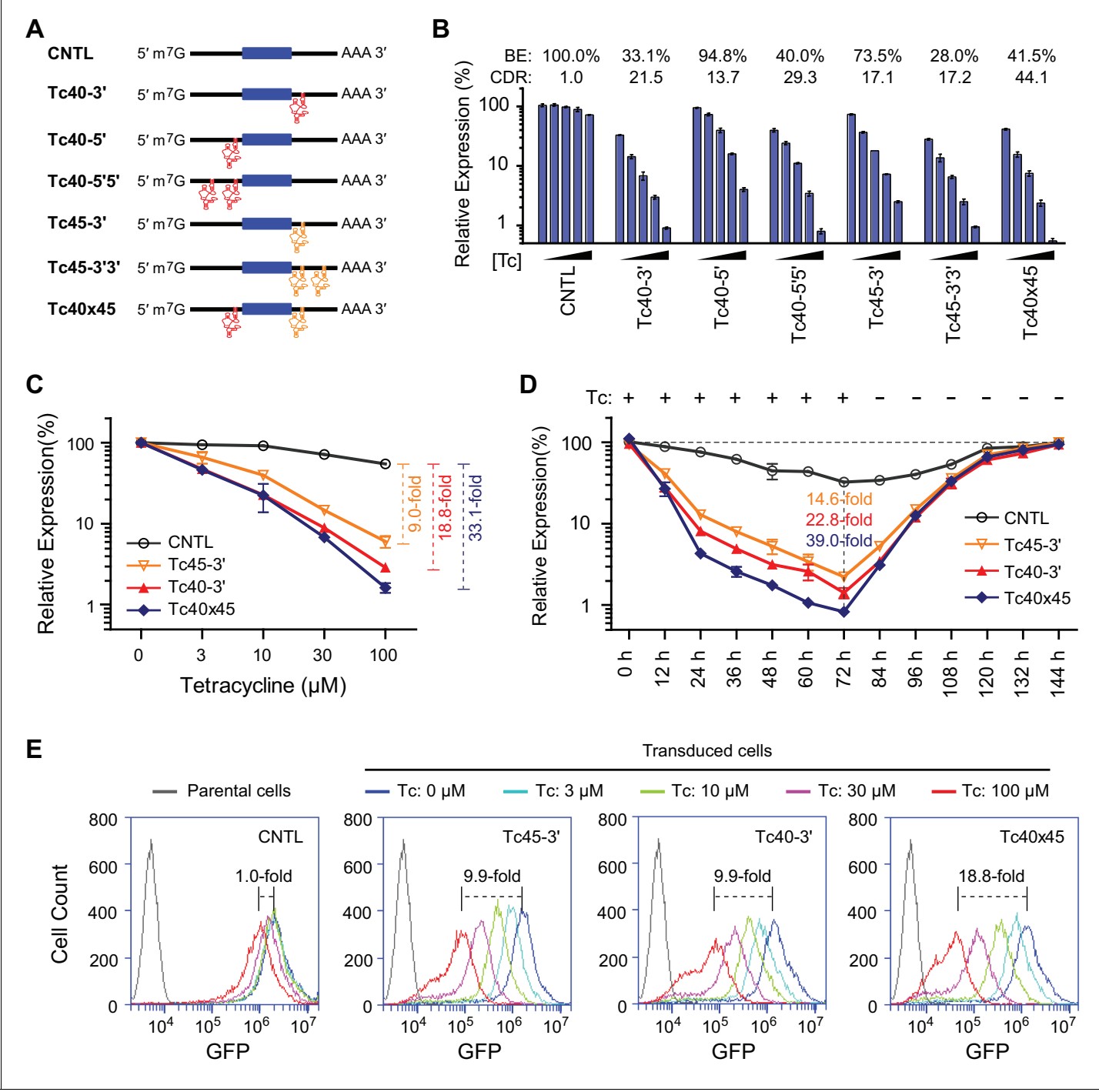

**Figure 4.** Control of retroviral vectored transgene expression using rationally designed Tc aptazymes. (**A**) A representation of the reporter constructs tested. Tc40 (red) or Tc45 (orange) were inserted either upstream (5') or downstream (3') of the GLuc gene (filled blue box), either as a single copy (Tc40-5', Tc40-3', or Tc45-3'), in tandem repeats (Tc40-5'5' or Tc45-3'3'), or across the Gluc gene (Tc40×45). (**B**) Plasmids containing the Tc variants represented in (**A**) were tested by transient transfection. Transfected HeLa cells were cultured with 0, 3, 10, 30, or 100 µM Tc for 2 days. Secreted GLuc in the culture supernatant was measured by a luminescence assay. Luciferase expression levels are normalized to that of the CNTL control construct. BE and CDR at 100 µM Tc are shown above the figure. (**C**) HeLa cells were transduced with murine-leukemia virus (MLV) vectors expressing GLuc genes regulated by the indicated Tc aptazyme variants and selected by puromycin. Stably transduced cells were cultured with the indicated concentrations of Tc for 48 hr. Secreted GLuc in the supernatants was measured by luminescence assay. Luciferase expression level was normalized to that observed without Tc. CDR at 100 µM Tc is indicated with brackets at the right. (**D**) Stable cells characterized in (**C**) were cultured in 100 µM Tc for 72 hr. Tc was then withdrawn and the cells were cultured for an additional 72 hr. Cell culture supernatants were collected and replaced with fresh medium every

*Figure 4 continued on next page*

*Figure 4 continued*

12 hr. Secreted GLuc in the supernatants was measured by luminescence assay. Luciferase expression levels at each time point were normalized to those observed before Tc was added. CDR observed at 72 hr for each aptazyme variant is indicated. (E) Assays similar to those in (C) except that a destabilized enhanced green fluorescent protein (GFP) was used as a reporter, and flow cytometry was used to measure expression. CDR at 100 µM Tc is indicated with brackets. Data shown are representative of two or three independent experiments. Data points in panels B–D represent mean ± S.D. of three biological replicates. Data points in panel E are representative of three biological replicates.

The following figure supplement is available for figure 4:

**Figure supplement 1.** Control of retroviral vectored transgene expression by rationally designed aptazymes.

We also introduced Tc40-3′, Tc45-3′, and Tc40×45 into an AAV vector expressing Gluc, where these aptazymes worked as effectively as in transient transfection or retroviral transduction studies (*Figure 5A*), suggesting that they may be useful to gene-therapy applications. In this context, Tc40-3′ worked nearly as effectively as Tc40×45. We therefore assayed the ability of Tc40-3′ to regulate AAV-mediated expression of human coagulation factor IX and etanercept (Enbrel), a fusion of a TNF-receptor ectodomain and the human IgG1 Fc domain (*Mohler et al., 1993*). Factor IX and etanercept proteins are currently used to treat hemophilia B and rheumatoid arthritis, respectively, and both of these biologics are being assessed as AAV transgenes in clinical trials (*Evans et al., 2008*; *Nathwani et al., 2014*). In both cases, their expression was dramatically reduced by 30 µM Tc (*Figure 5B and C*), a concentration that is well tolerated clinically and readily achievable with oral administration (*Agwuh and MacGowan, 2006*).

We further sought to determine whether AAV transgene expression could be regulated by these aptayzmes in vivo. We prepared high-titer recombinant AAV1 viruses carrying the GLuc gene regulated or not by Tc40×45. These two vectors were then injected into the gastrocnemius muscle of BALB/c mice. Two in vivo studies were performed. In the first, ten male mice were injected with a Tc40×45-regulated or control vector, and imaged for luciferase expression on day 10 post injection. Mice were then treated four days later with tetracycline (250 mg/kg) or phosphate-buffered saline (PBS) for four days, and again imaged. All mice injected with control or Tc40×45-regulated vectors exhibited robust luciferase expression in the gastrocnemius muscle, and Tc treatment significantly reduced luciferase expression in the Tc40×45 mice but not the control mice (*Figure 5—figure supplement 1*). In a second experiment, fourteen female mice were again injected with a Tc40×45-regulated or control vector. Two weeks after injection, the mice were treated for three days with tetracycline or PBS, and imaged (*Figure 5D*). Combining both studies, we obtained an average dynamic range of 6.9-fold in vivo (*Figure 5E*), lower than that observed in tissue culture cell lines. We speculate that this difference is due to the high metabolism of tetracycline in mice (*Sizemore et al., 2002*; *Agwuh and MacGowan, 2006*), and inefficient access of tetracycline to AAV1-transduced muscle cells (*Bocker et al., 1984*). Ribozyme based control of AAV-transgene expression in mice has been reported (*Yen et al., 2004*), however this switch control is possibly mediated by direct incorporation of the regulatory drug toyocamycin into the ribozyme RNA (*Yen et al., 2006*). Therefore, our data present the first example of aptazyme-mediated regulation in primary animal tissues in vivo.

## Discussion

Aptazymes have the potential to be useful in a number of important scientific and medical applications. For example, they afford simple and immediate regulation of gene expression in cell biology studies. In addition, aptazymes can be introduced by CRISPR/Cas9 into an untranslated region of an exon to exogenously control endogenous genes in their native contexts. They also have the potential to transform gene therapy, allowing controlled dosing of a biologic with a well tolerated drug (*Figure 5A–C*). In these applications, aptazymes have several key strengths. First, they are smaller than most other gene-regulation systems, typically less than 200 base pairs. They therefore can be easily included in gene therapy vectors such as AAV with limited space for additional genes. Second, in contrast to transcription factor-dependent systems like Tet-Off (*Gossen and Bujard, 1992*), aptazyme-mediated regulation does not require an immunogenic non-self protein. Third, aptazymes

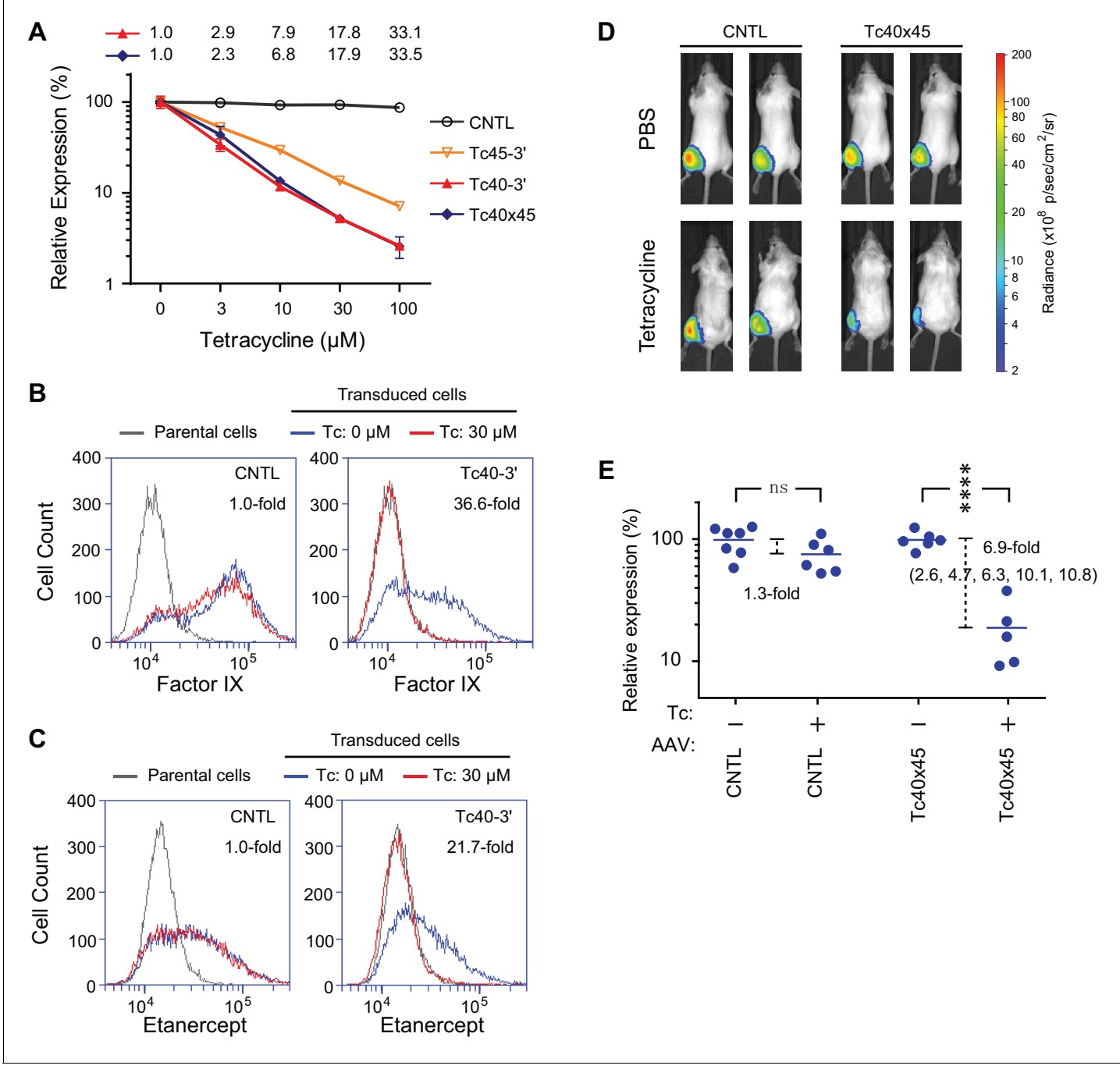

**Figure 5.** Aptazyme-mediated control of adeno-associated virus (AAV) vectored transgene expression. (**A**) HeLa cells were transduced with AAV vectors expressing GLuc regulated by the indicated aptazymes and were cultured with the indicated concentrations of Tc for two days post-transduction. Luciferase expression level was normalized to that observed without Tc. CDR for Tc40-3′ and Tc40×45 is indicated above the figure. (**B**) HeLa cells were transduced with an AAV vector expressing human coagulation factor IX regulated (right panel) or not (left panel) by Tc40-3′. Cells were then incubated in presence (red) or absence (blue) of 30 µM Tc for two days and analyzed by flow cytometry using intracellular staining with an anti-human factor IX antibody. Grey indicates parental HeLa cells stained with the same antibody. CDR is indicated in the upper right corner of the panels. (**C**) An experiment similar to that in (**B**) except that cells were transduced with an AAV vector expressing etanercept (Enbrel), and stained with an anti-human IgG1 Fc antibody. (**D**) Fourteen 9-week old female BALB/c mice were injected with $10^{11}$ genome copies of AAV particles carrying GLuc gene regulated (Tc40×45, seven mice) or not (CNTL, seven mice) by Tc40×45 into the left gastrocnemius muscle. All mice were treated for 3 days with Tc (250 mg/kg) or phosphate-buffered saline (PBS) two weeks post-injection, and imaged for luciferase expression using the Xenogen IVIS In-Vivo Imager. (**E**) Bioluminescent photon outputs of all the imaged mice were analyzed using the Living Image Software. Relative luciferase expression was calculated as the percentage of bioluminescent photon output relative to the average photon output of PBS-treated mice injected with same AAV vector. Each point

*Figure 5 continued on next page*

*Figure 5 continued*

represents one mouse, and bars indicate the average expression level for each treatment group. Average fold changes of the luciferase expression between Tc- and PBS-treated groups are indicated. Numbers in the parentheses are the fold regulation of individual Tc-treated mice injected with Tc40×45-regulated vector. Differences between Tc- and PBS-treated groups were significant (two-sided *t*-test, p<0.0001) in mice injected with the Tc40×45-regulated vector, whereas differences in the control mice were not significant (two-sided *t*-test, p>0.05). Data shown in (**A–D**) are representative of two or three independent experiments. Data points in panel **A** represent mean ± S.D. of three biological replicates. Data points in panel **B** and **C** are representative of three biological replicates.

The following figure supplement is available for figure 5:

**Figure supplement 1.** Control of AAV vectored transgene expression by rationally designed aptazymes.

cleave in *cis*, and only the message bearing an aptazyme is affected. They are therefore highly specific, with few off-target effects. Finally, as we show here, they can be tailored to different aptamers regulated by different ligands, including those shown to be safe in humans.

Here we describe a straightforward method for converting aptamers into aptazymes with wide dynamic ranges in mammalian cells. We began with several straightforward and intuitive principles. For example, a communication module that anneals with too high overall energy will induce premature auto-activation of the ribozyme, even when the aptamer is unbound. In contrast, a communication module that is unstable will not convey the state of the aptamer to the ribozyme. However, between these limiting cases, we found that the communication-module annealing energy correlated poorly with an aptazyme's dynamic range (*Figure 2A* and *Figure 2—figure supplement 1A and C*). As shown here, the reason for this poor correlation is that base pairs proximal to the ribozyme have a disproportionate impact on the ribozyme's enzymatic activity. Any energy-like function describing the impact of the communication module on the ribozyme should therefore take into account the distance of communication-module bases to ribozyme. Accordingly, we developed a scoring function, WHSS, based on the weighted sum of hydrogen bonds and inter-strand base-stacking interactions. This function accurately predicted the rank order of the BE of a test panel of 32 aptazymes (*Figure 2G* and *Figure 2—figure supplement 1F*), and was validated with 21 newly designed aptazymes (*Figure 3I*). Critically, when BE was plotted against WHSS, an inflection point was observed in the vicinity of a WHSS value of 6.7. As the WHSS values increased after this point, the BE of aptazymes rapidly decreased (*Figure 2G*). At this inflection point, then, additional energy provide by the ligand-bound aptamer has a greater impact on aptazyme activity and therefore its dynamic range (*Figure 2—figure supplement 2C and D*). Thus, in most cases, aptazymes with WHSS values near 6.7 exhibited relatively high CDRs (*Figure 2H and L*). We also observed that communication-module bases immediately adjacent to the aptamer affected CDR independently of WHSS (*Figure 2I–K*), likely due to their effect on ligand affinity.

Importantly, we show that this approach is general. For example, it has allowed us to develop tetracycline-regulated aptazymes with two different architectures. These aptazymes exhibited high levels of basal expression and dynamic ranges wider than any previously described aptazymes (*Wittmann and Suess, 2011*; *Beilstein et al., 2015*). To underscore the modularity of this approach, we also developed aptazymes regulated by two additional ligands, theophylline and guanine. In both cases, the newly developed aptazymes again significantly outperformed all previously described aptazymes (*Auslander et al., 2010*; *Nomura et al., 2012*). These data suggest that any aptamer which anneals an accessible stem region upon ligand binding would likely be convertible to an efficient aptazyme off-switch. Ligand-aptamer binding-induced stem formation or stabilization could also be used to develop aptazyme on-switches. For example, an on-switch could use ligand-induced stem formation or stabilization to hide the hammerhead-ribozyme loop II or bulge I sequences important to ribozyme activity (*De la Peña et al., 2003*; *Khvorova et al., 2003*; *Martick and Scott, 2006*; *Win and Smolke, 2007*; *Beilstein et al., 2015*). Our empirical approach may therefore be used to establish WHSS-like scoring functions for generating aptazyme on-switches. Thus this study provides a general rational-design method for rapidly converting diverse aptamers into aptazymes with wide regulatory ranges in mammalian cells.

# Materials and methods

## Plasmids

DNA fragments encoding aptazyme variants were synthesized by Integrated DNA Technologies (IDT, Coralville, IA). The reporter plasmids used in transient transfection and retroviral vector transduction assays were constructed by ligating a Gaussia luciferase (GLuc) or destabilized green fluorescent protein (GFP) reporter-gene and one or two aptazyme-encoding fragments into the pQCXIP vector plasmid (Clontech, Mountain View, CA) or pcDNA3.1(+) plasmid (Thermo Fisher Scientific, Waltham, MA). AAV vector plasmids bearing various aptazyme elements were constructed from the pAAV-MCS plasmid (Agilent Technologies, Santa Clara, CA).

## Cell culture

Human embryonic kidney 293T (RRID:CVCL_0063) and human cervical carcinoma HeLa (RRID:CVCL_0030) cells were maintained in Dulbecco's Modified Eagle Medium (DMEM, Life Technologies) at 37°C in a 5% $CO_2$-humidified incubator. All growth media were supplemented with 2 mM Glutamax-I (Life Technologies, Carlsbad, CA), 100 µM non-essential amino acids (Life Technologies), 100 U/mL penicillin and 100 µg/mL streptomycin (Life Technologies), and 10% tetracycline-free FBS (Clontech).

## Measurement of aptazyme-regulated gene expression in transiently transfected cells

HeLa or 293T cells seeded in 48-well plate with antibiotic-free growth medium were transfected with 6.25 ng of aptazyme-regulated GLuc expression plasmids using 0.3 µL Lipofectamine 2000 (Life Technologies). Three hours later, medium was removed and fresh growth medium containing 2% FBS was added. After an additional three hours, culture medium was changed to induction medium containing varying concentrations of tetracycline, theophylline, or guanine (Sigma-Aldrich, St. Louis, MO). Induction medium was replenished daily. One or two days after transfection, GLuc secreted into the supernatant was measured using a luminescence assay. All studies measuring the dynamic range of aptazyme variants were corrected for non-specific effects of tetracycline, theophylline, or guanine by dividing the ligand-induced fold change in aptazyme-regulated gene expression by the fold change observed in a control construct lacking any switch elements.

## Luminescence assay

To measure GLuc expression, 20 µL of cell culture supernatant of each sample and 100 µL of GLuc assay buffer were added to one well of a 96-well black opaque assay plate (Corning, Corning, NY), and measured with a multi-label plate reader (PerkinElmer, Waltham, MA). GLuc assay buffer consists of 7.5 mM sodium acetate, 250 mM sodium sulfate, 250 mM sodium chloride, and 4 µM coelenterazine native (Biosynth Chemistry & Biology, Staad, Switzerland) at pH 5.0.

## Calculation of communication-module scores

We calculated annealing energy of the communication modules ($\Delta G_{CM}$) using the online utility RNAstructure Web Server (*Bellaousov et al., 2013*). We also counted the number of potential hydrogen bonds (HBN) formed in the communication module. Both values correlated poorly with basal expression (BE). Specifically, Spearman's rank correlation coefficient (ρ) (*Zar, 1998*) was −0.64 for $\Delta G_{CM}$ and −0.80 for HBN (*Figure 2—figure supplement 1*). We observed that hydrogen bond interactions from every communication-module base pair contributed to basal expression (BE), but that the location of these hydrogen bonds relative to the ribozyme determined the scale of this contribution (*Figure 2B and C*). Accordingly, we assigned every communication-module base pair an empirically determined weight reflecting its proximity to the ribozyme, and multiplied this weight by the number of hydrogen bonds to assign a Weighted Hydrogen Bond Score (WHBS). Specifically, we observed a Spearman's rank correlation coefficient of −0.91 when the ribozyme-proximal starting base pair (A-U) was assigned a weight of 1.0, and the subsequent base pairs were sequentially assigned weights of 5/6, 4/6, 3/6, and 2/6 (*Figure 2—figure supplement 1*). When longer communication modules were considered, bases 6, 7, 8, and 9 were assigned weights of 1/6, 5/36, 4/36, and 3/36, respectively. We also observed that inter-strand purine base stacking in the communication

module also affected BE in a position-dependent manner (*Figure 2E and F*). We thus added a weighted sum of base-stacking interactions to the WHBS to calculate a Weighted Hydrogen-bond and Stacking Score (WHSS) for each aptazyme. The relative energy of a hydrogen bond and an RNA base-stacking interaction remains undefined (*Egli, 2009*; *Johnson et al., 2011*; *Šponer et al., 2012*). However, when potential inter-strand purine base-stacking interactions were assigned one half (G-G) or one quarter (A-A, A-G) of a hydrogen bond, we observed an optimal rank-order correlation ($\rho=-0.94$) between WHSS and BE (*Figure 2—figure supplement 1*). Calculated $\Delta G_{CM}$, HBN, WHBS, and WHSS values for all aptayzmes are presented in *Figure 2—source data 1* and *Figure 3—source data 1*.

## Production of retroviral and adeno-associated virus (AAV) vectors

293T cells plated at 70% confluence in T75 culture flasks were transfected using CalPhos transfection kit (Clontech) with different virus packaging plasmids. To produce pseudotyped murine leukemia viral (MLV) particles, cells were transfected with 10 μg of plasmid encoding the Machupo virus (MACV) envelope glycoprotein GP, 15 μg of plasmid encoding MLV Gag and Pol proteins, and 15 μg of pQCXIP-based plasmids encoding various aptazyme-regulated GFP or GLuc genes. Forty-eight hours after transfection, culture supernatants were harvested and filtered through 0.45 μm filters. AAV particles were produced by transfecting cells with 13 μg of plasmid encoding AAV replication and capsid proteins, 13 μg of plasmid encoding helper proteins, and 13 μg of pAAV-MCS-based plasmids encoding aptazyme-regulated GLuc, etanercept, or factor IX. Sixty to seventy-two hours post-transfection, culture supernatants were harvested and filtered through 0.45 μm filters.

## Measurement of aptazyme-regulated gene expression in stable cells

HeLa cells transduced with pseudotyped MLV were selected with growth medium containing 4 μg/mL puromycin (Life Technologies) to generate aptazyme-regulated GLuc- or GFP-stable cells. Stable cells were then plated in 96-well plates to assay expression of GLuc or GFP in the presence of varying concentrations of tetracycline. Two days later, reporter expression was measured using luminescence assay or fluorescence microscopy. For time-course experiments, GLuc stable cells were cultured in the presence of 100 μM tetracycline for 3 days, and in the absence of tetracycline for an additional 3 days. Every 12 hr, supernatant was removed and replaced with fresh medium and GLuc expression was measured.

## Measurement of aptazyme-regulated gene expression in AAV transduced cells

HeLa cells plated in 96-well plates were transduced with $1\times10^9$ genome copies of AAV particles. Four hours post-transduction, viral supernatant was removed and growth medium containing 2% FBS was added. Culture medium was replaced with induction medium containing varying concentrations of tetracycline 8 and 16 hr post-transduction. Two days post-transduction, target gene expression was measured using luminescence assay or flow cytometry.

## Immunofluorescence staining

To measure etanercept or human coagulation factor IX protein expression, AAV- or mock-transduced cells were trypsinized, fixed with 1% paraformaldehyde (Sigma) for 30 min on ice, permeabilized and blocked for 30 min on ice with a buffer containing 0.1% saponin (Sigma) and 3% bovine serum albumin (bioWORLD, Irving, TX). For etanercept staining, cells were incubated with 5 μg/mL Alexa Fluor 488-conjugated goat anti-human IgG Fc (Jackson ImmunoResearch Labs, West Grove, PA; Cat. No. 109545098, RRID:AB_2337840) for 30 min at room temperature. For factor IX staining, cells were sequentially incubated with 5 μg/mL mouse anti-human factor IX monoclonal antibody, (Haematologic Technologies Inc, Cat. No. AHIX-5041, RRID:AB_2629498) for 1 hr at room temperature, and 5 μg/mL Alexa Fluor 488-conjugated goat anti-mouse IgG (Jackson ImmunoResearch Labs, Cat. No. 115545071, RRID:AB_2338847) for 30 min at room temperature. Stained cells were analyzed using BD Accuri flow cytometer and BD CSampler software (BD Biosciences, San Jose, CA). Mock-transduced cells stained with corresponding antibodies were used as negative controls.

## Measurement of aptazyme-regulated gene expression in mice

Twelve 5-week old male and sixteen 9-week old female BALB/c mice (Charles River Laboratories, strain code 028, RRID: MGI:2683685) were randomly separated into weight-matched groups without blinding, and injected in the left gastrocnemius muscle with 20 µL of purified AAV particles ($3 \times 10^{10}$ or $10^{11}$ genome copies per mice) carrying GLuc gene regulated or not by Tc40×45 switch elements. Ten days post-injection, male mice were imaged for luciferase expression using the Xenogen IVIS In-Vivo Imager (PerkinElmer). Two weeks post-injection, mice were intraperitoneally injected with tetracycline at 250 mg/kg per day or with phosphate-buffered saline (PBS) for three (female) or four (male) days. Two male mice and two female mice that rapidly lost more than 15% of their body weight were eliminated from the study. The remaining ten male mice and fourteen female mice were then imaged for luciferase expression using the In-Vivo Imager.

## In vivo bioluminescent imaging

Prior to imaging, the mice were anesthetized with isoflurane, then intramuscularly injected with 100 µg of 'Inject-A-Lume' *Gaussia* luciferase substrate (Nanolight Technology, Pinetop, AZ), and returned to their cages for recovery. Fifteen minutes after substrate injection, mice were anesthetized again and imaged for bioluminescence using the IVIS In-Vivo Imager (Xenogen, Alameda, CA). Bioluminescent photon outputs of different images were normalized and quantified using the Living Image Software. Tc-induced fold changes of the luciferase expression were calculated by dividing the photon output of each Tc-treated mouse by the average photon output of PBS-treated mice injected with same vector. Two-sided *t*-test was performed to analyze the statistical significance of the expression difference between Tc- and PBS-treated groups. All mice studies were performed in accordance with the Scripps Florida Institutional Animal Care and Use Committee animal use protocol number 14–028.

## Acknowledgements

The authors would like to thank Albert Anbo Zhong for careful reading of the manuscript and insightful comments.

## Additional information

### Funding

| Funder | Grant reference number | Author |
| --- | --- | --- |
| National Institutes of Health | R01 AI091476 | Michael Farzan |
| National Institutes of Health | P01 AI100263 | Michael Farzan |

The funders had no role in study design, data collection and interpretation, or the decision to submit the work for publication.

### Author contributions

GZ, Conception and design, Acquisition of data, Analysis and interpretation of data, Drafting or revising the article, Contributed unpublished essential data or reagents; HW, Acquisition of data, Drafting or revising the article, Contributed unpublished essential data or reagents; CCB, Acquisition of data, Drafting or revising the article; GG, Drafting or revising the article, Contributed unpublished essential data or reagents; MF, Conception and design, Analysis and interpretation of data, Drafting or revising the article, Contributed unpublished essential data or reagents

### Author ORCIDs

Michael Farzan, http://orcid.org/0000-0002-2990-5319

### Ethics

Animal experimentation: This study was performed in strict accordance with the recommendations in the Guide for the Care and Use of Laboratory Animals of the National Institutes of Health. All of

the animals were handled according to approved institutional animal care and use committee (IACUC) protocol (#14-028) of the Scripps Florida.

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
