## [Decision Letter]

Thank you for submitting your article "Rational design of aptazyme riboswitches for efficient control of gene expression in mammalian cells" for consideration by *eLife*. Your article has been favorably evaluated by Randy Schekman (Senior Editor) and three reviewers, one of whom, Ronald Breaker (Reviewer #1) served as Guest Editor.

The reviewers have discussed the reviews with one another and the Reviewing Editor has drafted this decision to help you prepare a revised submission.

Summary:

Dr. Farzan and coworkers have presented a general method for the rapid and efficient development of ligand-responsive RNA genetic switches that can be used to regulate gene expression in mammalian cells. The authors use a novel intellectual framework to guide the engineering of unique linker sequences or communication modules between ligand-binding aptamer domains and self-cleaving ribozymes. By synthesizing and testing a series of RNA constructs, the authors generate a set of empirical data that reveals the importance of hydrogen bond number, base stacking, and distance between key RNA structures to functional switch outcomes. For successful switches, the binding of a corresponding ligand to the aptamer domain alters ribozyme function at a level sufficient to substantially affect expression from mRNAs carrying the engineered RNAs. The engineering approach is demonstrated to apply to multiple different aptamer domains, suggesting that the design principles be applicable to a large diversity of additional RNA domains. The concepts and methods presented in this work should permit others to create novel types of engineered riboswitches for practical applications.

Essential revisions:

1) The authors should consider making a more appropriate comparison between their RNA switches and the mammalian Tc switches created by Suess and coworkers.

*Reviewer #1:*

This manuscript by Zhong et al. and Farzan provides convincing support for an exciting strategy to create RNA switches or 'aptazyme riboswitches' that work inside mammalian cells. The ability to control gene expression in a precise manner has been a long-sought goal by a number of laboratories, as various applications can be envisioned. As examples, the capability to create new switches quickly would allow researchers to more easily artificially regulate, and thereby study, numerous biological systems, and synthetic biologists would have a versatile set of gene control tools with which to create new regulatory networks. Moreover, there will be opportunities to use engineered RNA switches for future gene therapy applications.

In the current study, the authors have used an empirical approach to create an apparently powerful intellectual framework that yields novel switches that respond to several small molecules. Using their "Weighted Hydrogen-bond and Stacking Score" or WHSS, they demonstrate that each communication module made exhibits a predictable value for basal gene expression, ligand-inhibited expression, and corrected dynamic range. Perhaps future researchers will add additional biophysical rigor to understand precisely how this approach works at atomic resolution. However, it seems very likely that the authors have made an excellent technical advance for this field, and I do not think this work should suffer publication delays until all this is better understood. The success of their WHSS system plus the ad hoc retention of the aptamer-proximal base-pair is remarkable. Simply put, it seems likely that other researchers can easily adopt the strategy described by the authors to create novel RNA switches that work well in cells. I have only two very minor comments:

1. In Figure 2 and elsewhere, the authors should consider annotating a 5' terminus of the sequences depicted just to ensure that the readers maintain the right orientation.

2. The references are not entirely in the style of the journal and these should be edited accordingly.

*Reviewer #2:*

Farzan and co-workers present a novel algorithm for the rational design of communication modules of ligand-responsive ribozymes. It takes into account additional parameters that allow for designing switches with better performances. The obtained responses with tetracyclin-dependent HHR systems in human cell culture are very impressive. Moreover, they demonstrate that the approach can be extended to theophylline- and guanine-dependent systems and that in these cases also improved performances are obtained. In addition, they demonstrate the application of tetracycline-dependent switches in AAV-mediated transgene expression in mice. This part is especially interesting since it demonstrates a principle applicability of aptazymes as switches in gene therapy applications. Taken together I can support acceptance for *eLife* given that the following (in some cases major) issues are appropriately addressed:

In general, the comparison of performances of artificial riboswitches is difficult because different expression systems etc. are often compared. On the other hand, in order to demonstrate the power of the approach such comparisons are necessary. However, in the present case the authors compare their Tc switches with the performance of an HHR-based switch by Wittmann and Suess. This switch has been engineered to function in yeast, however for comparison it is utilized in mammalian cells. It is reported that the switch is not working under these conditions; this is also why Suess and coworkers engineered Tc switches in mammalian cells which function very nicely as ON-switches. Hence the comparison is not valid and without further comment very unjustified sheds doubt on the work of Suess and co-workers. Why not compare the ON-switches with the present Tc switches? It is the dynamic range that defines the performance, even if the switching mode is opposite.

Also, regarding the comparison of performances, two more ribozyme-based switches are compared (guanine and theophylline). More background information should be given here. For example, it is unclear how exactly these previously reported ribozymes were tested. Sequences and how they were inserted into the reporter mRNAs should be provided. Or are the values shown in Figure 3 adopted from the literature?

Measurement of gene expression: When quantifying gene expression in mammalian cell culture, it is good practice to use dual luciferase systems in order to correct for transfection efficiency etc. The authors present all their results using just one reporter without a suited control. Is this a problem? To be discussed with the other reviewers.

Limitation of the design algorithm: The algorithm seems to work nicely, however it only allows designing OFF-switches since it works via computing solutions for stabilization of the aptazyme structure upon ligand binding. The design of ON-switches required other designs. This aspect should at least be discussed.

The AAV-mediated transgene expression in mice is very impressive; however, ligand-dependent ribozymes have been utilized before in mice for controlling gene expression by Mulligan and co-workers. Although that work is limited and the present work extends this concept very nicely, the mentioned work needs to be discussed appropriately. The respective publication (Yen et al., Nature, 2004) is mentioned briefly in the Introduction and later in a technical context but it is not discussed with regard to transgene expression in a mammalian organism.

In the last sentence, the authors state that "Thus this study provides a robust rational-design method for rapidly converting diverse aptamers into aptazymes with wide regulatory ranges in mammalian cells." Since no new ligand selectivity has been demonstrated, this sentence is too optimistic. Only three known ligand-responsive systems have been optimized.

*Reviewer #3:*

Zhong et al. describe a clear and systematic study describing the engineering of aptamers into aptazymes to modulate gene expression in mammalian cells. This work builds off seminal papers describing riboswitches and aptazymes (Breaker lab) and earlier engineering of hammerhead ribozymes in mammalian cells (Mulligan lab) to establish important design principles across the different RNA modules (i.e., communication module, aptamer, ribozyme) to quantitatively characterize the key determinants that govern basal expression, ligand inhibition and dynamic range of their engineered aptazymes. Specifically, the authors elucidate the impact of the number of hydrogen bonds (as opposed to δ-G) and the distance of communication-module bases to ribozyme have on basal expression and dynamic range of the aptazyme switches. This allowed the authors to develop a scoring function (WHSS) to predict the basal expression of 32 aptazymes that was subsequently validated with 21 newly designed aptazymes. Notably, the authors applied their findings to develop aptazymes of different architectures and for two additional ligands (theophylline and guanine). The systematic and exhaustive suite of aptazyme variants coupled by their detailed characterization is an important strength of this manuscript and a necessary and welcome step in this field. Finally, the authors describe an impressive demonstration of aptazyme-mediated gene regulation in a mouse model. Overall, I think the paper is clear and describes a rigorous set of experiments to support the key results and conclusions of the manuscript. I only have a few minor points that I encourage the authors to pursue before publication in *eLife*:

1. Subsection “Validation of design principles through development of four classes of efficient aptazymes” and Figure 3 – dynamic range for CDR seems arbitrarily defined. I encourage the authors to establish quantitative bounds for low-medium-high CDR. I also think the authors have a typo – did they intend to refer to Tc44 rather than Tc43?

2. In the first paragraph of the Discussion, the authors propose introducing aptazymes via CRISPR/cas9 into precise positions in the genome. Given some of the specificity constraints via PAM sites and the potential for off-target affects or mutagenic activity of NHEJ, do the authors have a citation that reports such a capability or their own data to support this statement? Achieving this level of precision via CRISPR/cas9 could be quite challenging and worthy pursuit.

3. To what extent was the dosing required to achieve the observed (6.9-fold) switch-like behavior toxic to the animal? Commenting on this would be helpful in actualizing some of the discussion points associated with safe gene therapy applications currently described in the first paragraph of the Discussion.

4. In my opinion, I recommend changing the order of the Discussion points. Specifically, many points in the first paragraph are better suited in the last paragraph of the Discussion and preceded by discussion points (currently second-third paragraphs of the Discussion) that are more central to the key findings of this study.

5. Although the authors demonstrate the ability to engineer aptazyme behavior with three commonly used ligands and associated aptamers, I think it would strengthen the paper and the authors' claims of broad applicability for the authors to comment on other ligand-aptamer pairs that they attempted and similarly how they envision adapting their programmable strategy for other ligands that may not yet have existing aptamers, highlighting challenges.

---

## [Author Response]

*[…] Essential revisions:*

*1) The authors should consider making a more appropriate comparison between their RNA switches and the mammalian Tc switches created by Suess and coworkers.*

The tetracycline (Tc) aptazyme off-switch 9.19 (Wittmann and Suess, 2011) created by Suess and coworkers was originally developed by in vitro selection. They then tested some candidates for control of reporter gene expression in yeast as well as in mammalian cells. One candidate, 9.19, showed around 2-fold dynamic range in yeast but no any regulation in mammalian cells. As 9.19 is the only reported Tc aptazyme off switch functional in eukaryotic cells, we chose it for the comparison in Figure 3.

Suess and coworkers have also engineered Tc aptazyme on switches functional in mammalian cells, and their best construct, 3'K19, showed 4.8-fold dynamic range at 50 µM Tc and 8.7-fold dynamic range at 250 µM Tc (Beilstein et al., 2015). Although we did not originally include a comparison with 3'K19 because the switching mode was opposite to that of our off switches, we have now included a direct comparison (Figure 3).

For the reviewers’ benefit, we provide more technical details. Specifically, we did a careful comparison between our Tc40 construct and 3'K19. We cloned both into 3'-UTR of a *Gaussia* luciferase (GLuc) gene in pcDNA3.1(+) plasmid. Both were flanked with same spacer sequences (Win and Smolke, 2007) to avoid interference of aptazyme folding by the surrounding sequences, and inserted between *Bam*HI and *Eco*RI restriction sites immediately downstream of the GLuc stop codon. They were then transfected into HeLa cells for measurement of corrected dynamic range (CDR) in response to different concentrations of Tc. The CDR data and relative GLuc expression data were presented in Figure 3 and Figure 3—figure supplement 2. Consistent with our previous data with Tc40 in different plasmid context, as well as that reported by Beilstein et al., Tc40 (19.6-fold at 100 µM Tc) showed significantly better CDR than that of 3'K19 (6.6-fold at 100 µM Tc).

*Reviewer #1:*

*This manuscript by Zhong et al. and Farzan provides convincing support for an exciting strategy to create RNA switches or 'aptazyme riboswitches' that work inside mammalian cells. The ability to control gene expression in a precise manner has been a long-sought goal by a number of laboratories, as various applications can be envisioned. As examples, the capability to create new switches quickly would allow researchers to more easily artificially regulate, and thereby study, numerous biological systems, and synthetic biologists would have a versatile set of gene control tools with which to create new regulatory networks. Moreover, there will be opportunities to use engineered RNA switches for future gene therapy applications.*

In the current study, the authors have used an empirical approach to create an apparently powerful intellectual framework that yields novel switches that respond to several small molecules. Using their "Weighted Hydrogen-bond and Stacking Score" or WHSS, they demonstrate that each communication module made exhibits a predictable value for basal gene expression, ligand-inhibited expression, and corrected dynamic range. Perhaps future researchers will add additional biophysical rigor to understand precisely how this approach works at atomic resolution. However, it seems very likely that the authors have made an excellent technical advance for this field, and I do not think this work should suffer publication delays until all this is better understood. The success of their WHSS system plus the ad hoc retention of the aptamer-proximal base-pair is remarkable. Simply put, it seems likely that other researchers can easily adopt the strategy described by the authors to create novel RNA switches that work well in cells.

We thank the reviewer for his thoughtful and generous observations. We have addressed the minor points raised, as described above.

*Reviewer #2:*

*[…] In the last sentence, the authors state that "Thus this study provides a robust rational-design method for rapidly converting diverse aptamers into aptazymes with wide regulatory ranges in mammalian cells." Since no new ligand selectivity has been demonstrated, this sentence is too optimistic. Only three known ligand-responsive systems have been optimized.*

The reviewer makes several points well worth addressing. First, he/she notes that comparisons among aptazymes are challenging or even unfair because switches function optimally in different contexts (e.g. yeast vs. mammalian cells). Also, we did not include a comparison with a very useful on switch that functions nicely in mammalian cells. We have now addressed this by including such a comparison (Figure 3). We would also note that our quantitative results conform nicely to descriptions of both on- and off-switches published by the Suess lab. To address a question of this reviewer, Figure 3 were experiments done in the lab, not taken from the literature. We have included GLuc relative expression data supporting Figure 3 in Figure 3—figure supplement 2. Finally, at the suggestions of the reviewer, we have now discussed how our approach might be adapted to on-switch development, discussed AAV-regulation described in a paper from the Mulligan group (Yen et al., 2004), and toned down the optimism of our final sentence.

*Reviewer #3:*

*[…] Overall, I think the paper is clear and describes a rigorous set of experiments to support the key results and conclusions of the manuscript. I only have a few minor points that I encourage the authors to pursue before publication in eLife:*

*1. Subsection “Validation of design principles through development of four classes of efficient aptazymes” and Figure 3 – dynamic range for CDR seems arbitrarily defined. I encourage the authors to establish quantitative bounds for low-medium-high CDR. I also think the authors have a typo – did they intend to refer to Tc44 rather than Tc43?*

*2. In the first paragraph of the Discussion, the authors propose introducing aptazymes via CRISPR/cas9 into precise positions in the genome. Given some of the specificity constraints via PAM sites and the potential for off-target affects or mutagenic activity of NHEJ, do the authors have a citation that reports such a capability or their own data to support this statement? Achieving this level of precision via CRISPR/cas9 could be quite challenging and worthy pursuit.*

*3. To what extent was the dosing required to achieve the observed (6.9-fold) switch-like behavior toxic to the animal? Commenting on this would be helpful in actualizing some of the discussion points associated with safe gene therapy applications currently described in the first paragraph of the Discussion.*

*4. In my opinion, I recommend changing the order of the Discussion points. Specifically, many points in the first paragraph are better suited in the last paragraph of the Discussion and preceded by discussion points (currently second-third paragraphs of the Discussion) that are more central to the key findings of this study.*

*5. Although the authors demonstrate the ability to engineer aptazyme behavior with three commonly used ligands and associated aptamers, I think it would strengthen the paper and the authors' claims of broad applicability for the authors to comment on other ligand-aptamer pairs that they attempted and similarly how they envision adapting their programmable strategy for other ligands that may not yet have existing aptamers, highlighting challenges.*

We thank the reviewer for his/her generous comments. We have now included quantitative bounds for our CDR, and corrected the error noted by the reviewer. Given the explosive changes with CRISPR/Cas9, we have not modified the text as suggested, and note here that Cas9 and Cpf1 have been engineered or identified with novel PAM specificities, and we think of our comments regarding CRISPR systems as somewhat forward looking. With regard to toxicity in our in vivo studies: as indicated in the methods, four out of 28 mice exhibited some negative reaction to tetracycline treatment, as indicated by weight loss. These mice were removed from the study and not analyzed further. No outward signs of toxicity were observed in the remaining mice. Finally, we have not yet attempted to engineer aptazymes with ligand different from the three described in the paper.